# Influence of Air-Drying Conditions on Quality, Bioactive Composition and Sensorial Attributes of Sweet Potato Chips

**DOI:** 10.3390/foods12061198

**Published:** 2023-03-12

**Authors:** Elsa M. Gonçalves, Nelson Pereira, Mafalda Silva, Nuno Alvarenga, Ana Cristina Ramos, Carla Alegria, Marta Abreu

**Affiliations:** 1UTI—Unidade de Tecnologia e Inovação, Instituto Nacional de Investigação Veterinária e Agrária, Av. da República, Quinta do Marquês, 2780-157 Oeiras, Portugal; 2GeoBiotec—GeoBioTec Research Institute, Faculdade de Ciências e Tecnologia, Universidade Nova de Lisboa, Campus da Caparica, 2829-516 Caparica, Portugal; 3Faculdade de Ciências e Tecnologia, Universidade Nova de Lisboa, Campus da Caparica, 2829-516 Caparica, Portugal; 4cE3c—Centre for Ecology, Evolution and Environmental Changes & CHANGE—Global Change and Sustainability Institute, Faculdade de Ciências, Universidade de Lisboa, 1749-016 Lisbon, Portugal; 5LEAF—Linking Landscape, Environment, Agriculture and Food Research Center, Associated Laboratory TERRA, Instituto Superior de Agronomia, Universidade de Lisboa, Tapada da Ajuda, 1349-017 Lisbon, Portugal

**Keywords:** sweet potato chips, drying, total phenolic content, total carotenoid content, sensory evaluation

## Abstract

The drying process is an essential thermal process for preserving vegetables and can be used in developing dried products as healthy alternative snacks. The effects of air-drying conditions using a convection dryer with hot air at different temperatures (60°, 65°, 70°, 75°, and 80 °C, in the range 5–200 min, at a fixed air speed of 2.3 m/s) were tested on the quality of slices (2.0 ± 0.1 mm) of dried sweet potato (Bellevue PBR). For each time and temperature, drying condition, physicochemical parameters (moisture content, CIELab color, texture parameters, total phenolic and carotenoid contents) and a sensory evaluation by a panel at the last drying period (200 min) were assessed. Drying time was shown to have a more significant effect than temperature on the quality of dried sweet potato as a snack, except for carotenoid content. Given the raw tuber content, thermal degradation (*p* < 0.05) of total phenolic compounds (about 70%), regardless of tested conditions, contrasted with the higher stability of total carotenoids (<30%). The dried product, under optimal conditions (≥75 °C for 200 min), achieved a moisture content (≤10%) suitable for preservation, providing a crispy texture with favourable sensory acceptance and providing a carotenoid content similar to the raw product.

## 1. Introduction

Sweet potato (SP) (*Ipomoea batatas* L.) is a dicotyledonous plant which belongs to the Convolvulaceae family that produces edible tuberous roots. SP presents wide variations in shape, weight, and color of pulps(e.g., orange, purple, or yellowish-white), corresponding to the different carotenoid content, depending on cultivar and environmental conditions. Its consumption has prompted interest due to its high nutritional value (source β-carotene, provitamin A), high levels of antioxidants, richness in fibers, micronutrients, and since it is an excellent energy source, associated with a low glycemic index for the consumer [1].

According to data made available by the FAO Statistical Database, in 2019, more than 92 million tons (MT) of sweet potatoes were produced worldwide, with a harvest area of around 8 million hectares, Asia being the primary region of SP cultivation with a production of 59 MT [2]. The European sweet potato market has not yet reached its full potential. Growing imports and local production initiatives foster mainstream consumption and product development. Portugal’s SP production increased by 40% from 2012 to 2016, according to FAOSTAT [2], reflecting this “new” food’s increased popularity and demand for it.

The SP variety Bellevue PBR is distinguished by its orange flesh and smooth copper skin. This SP variety also has a superior root shape, storage capacity, and disease resistance, making it one of the most popular.

The uses of SP include the production of starches, flour, and purees. Like traditional potato chips, fresh and frozen sweet potato fries are popular products [3,4]. However, fried chips might not be the healthier vegetable snack choice. Instead, like the raw plant, dehydrated vegetable chips can serve as healthy snacks once they are nutritionally interesting. This kind of product is a current dietary recommendation in different countries, intending to help control hunger and obtain all the benefits of health-promoting essential nutrients, with a desirable flavor and aroma from the products that give origin to it [5].

Drying is one of the most common methods of food preservation. The drying process is a unitary operation defined as removing moisture by exposing the food to a heat source. As a result, less free water reduces microbial growth and biochemical reactions, consequently increasing food shelf life. Hot air drying is often used for fruit and vegetable products. It is a relatively simple and cheap food preservation method compared to other methods, such as ultrasound-assisted hot air or infrared drying methods, making it a popular choice for small-scale food processing operations [6].

Despite these advantages, the products’ physicochemical and organoleptic characteristics can be adversely altered [7] due to the product type, drying method, and time–temperature regime applied [8]. Applying a high drying temperature can reduce drying time but may involve severe damage to the product. On the other hand, water vaporization can also cause damage to the product, possibly leading to structural collapse [7]. Indeed, with no adequate control, there are changes in flavor, shrinkage and degradation of nutritional compounds (e.g., bioactive compounds) [9]. The effect of the drying process on vegetable quality was evaluated by different authors on products such as cabbage [10], potato [11] and sea cucumber [12]. Singh et al. [13] studied the effect of air temperature (50–60 °C), dehydration, and pretreatments on sweet potato cubes, and Onwude et al. [14] monitored and predicted the quality properties of sweet potatoes during drying using a computer vision imaging system. However, studies on sweet potato chips are limited to evaluating different cooking methods, such as frying and air frying [15,16]. Therefore, the main objective of this study was to investigate a cost-efficient pipeline alternative to support small-scale sweet potato production in Portugal. In accordance, the effects of various drying conditions with a hot air convection dryer on color, texture, phenolic content, and carotenoid content were assessed in the development of high-quality sweet potato crispy snacks, a convenient and healthy alternative with attractive sensory appeal.

## 2. Materials and Methods

### 2.1. Plant Material and Drying Processes

The orange-fleshed ‘Bellevue’ sweet potato [*Ipomoea batatas* (L.) Lam.] used in this study was supplied by Nativaland Lda, a company aiming to support farmers in increasing their productivity, quality and crop diversity based in Portugal.

On arrival at the laboratory, the sweet potatoes were hand selected regarding uniformity of size, appearance and absence of external damage. After cleaning to remove the dust with water, three batches of sweet potatoes (SP) of 10 kg each were stored in a handling warehouse (12.5 ± 1 °C, 85% RH) until processed. Before drying, the SPs were cut into 2.0 ± 0.1 mm thick slices in the transversal direction with a vegetable cutter (CL 50, Robot Coupe, France). The samples (±250 g) were placed in a forced-air tray dryer (Computer Controlled Tray Dryer, Armfield, England), and dehydration was performed at an air velocity of 2 m/s and at five different temperatures (60°, 65°, 70°, 75° and 80 °C). Mass loss (ML) was monitored until samples reached a constant mass value. During dehydration, sample weight was monitored every 3 min using an in-built digital balance (Mettler-Toledo PE 360, Switzerland) and samples were taken for analysis at different drying times (0, 5, 20, 60, 120 and 200 min). This procedure was replicated three times for each SP batch received. The samples and the raw plant were characterized according to the protocol described in Section 2.2 and Section 2.3. The initial raw material characterization is shown in Table 1.

### 2.2. Physicochemical Analysis

The pH was measured by potentiometry (Crison Micro pH 2001, Crison Instruments, Barcelona, Spain) in H_2_O using a 1:2 SP weight: extract-volume ratio. Water activity was determined by a digital hygrometer (Hygrolab 2, Rotronic AG., Bassersdorf, Switzerland). Soluble Solids Content (SSC, °Brix) was measured using an Atago DDR-A1 refractometer (ATAGO Co Ltd., Tokyo, Japan). Moisture content (MC) and titratable acidity (g citric acid/100 g product) were measured using NP-875 [17] and NP-1421 [18], respectively. The analyses were performed in triplicate per each batch.

An adaptation of the method described by Aprajeeta et al. [19] was used for SP shrinkage evaluation during drying. The shrinkage value was obtained by the ratio between each slice diameter at a determined time (D) and the diameter of the original sample (D0), giving the result as a percentage. SP slices with standardized dimensions (3.5 ± 0.3 cm diameter; 2.0 ± 0.1 mm thick) were used for shrinkage value determination. The diameter was obtained by averaging 16 chip slices of each tested time-temperature condition. A photographic recording was performed with an opaque black background (black cardstock). Full details of the equipment used can be found in Dias et al. [20].

A colorimeter (Minolta CR-300, Osaka, Japan) was used to evaluate the samples’ color by measuring CIEL*a*b* parameters (C illuminant, second observer). The instrument was calibrated using a white tile standard (L* = 97.10; a* = 0.08; b* = 1.80). The measurements were performed on the surface of ten sweet potato slices for each condition per batch. Chroma index (C*), hue index (°h), whiteness index (WI) and total color difference (ΔE) were calculated as follows:(1)$$C^{*}=\sqrt{\left(a^{*2}+b^{*2}\right)}$$
(2)$${}^{\circ}h=\tan^{-1}\frac{b^{*}}{a^{*}}$$
(3)$$\mathrm{WI}=100-\sqrt{{\left(100-L^{*}\right)}^{2}+a^{*}{}^{2}+b^{*}{}^{2}}$$
(4)$$\Delta E=\sqrt{{\left(\Delta L^{*}\right)}^{2}+{\left(\Delta a^{*}\right)}^{2}+{\left(\Delta b^{*}\right)}^{2}}$$
where ΔL*, Δa*, Δb* represent the deviations of the test and raw samples.

Analyses of the SP slice’s texture were performed using a texture analyzer (Stable Micro-System Ltd., Godalming, UK). The analysis was performed in 10 slices, with two perforations in each sample per batch. The conditions were: puncture test, 2 mm diameter cylindrical probe, 10 mm penetration, 1 mm/s test speed and 5 mm/s posttest speed. The parameters obtained from the Force–Distance curves were: hardness (H; N), breaking distance (BD; mm), crispness (CP; N/mm), work of fracture (W; N.mm) [21].

### 2.3. Phytochemical Analysis

For total phenolic content (TPC) determination, SP tissue was homogenized with methanol (1:4, w:v) and left overnight at 5 °C. Homogenates were centrifuged at 29,000 g for 15 min at 5 °C (Sorvall RC5C, rotor SS34, Sorvall Instruments, Du Pont, Wilmington, DE, USA), and the clear supernatant (methanolic extract) was used for total phenolic content determination using the method described by Swain and Hillis [22]. Methanolic extracts (150 μL) were diluted with nano-pure water (2400 μL) in test tubes, followed by the addition of 0.25 N Folin-Ciocalteu reagent (150 μL). The mixture was incubated for 3 min, and 300 μL of 1 N Na_2_CO_3_ was added. The final mixture was incubated for 2 h at room temperature in the dark. Spectrophotometric readings at 725 nm were collected using a JASCO V-530 UV/VIS spectrophotometer (Jasco International, Tokyo, Japan). The TPC results were expressed as milligrams of gallic acid equivalents per 100 g (mg GAE/100 g) of SP samples (dry weight basis), interpolated from a standard curve developed for this standard (0–0.35 mg/mL).

Total carotenoid content was determined according to Talcott and Howard [23]. In semi-dark conditions, SP sample was mixed (1:10, *w*:*v*) with an acetone/ethanol (1:1) solution containing 200 mg/L butylated hydroxytoluene (BHT). The mixture was homogenized until uniform consistency. The homogenate was filtered through a Whatman #4 filter, and further washes with acetone/ethanol solvent were followed until no further color change was observed and to a final volume of 100 mL. The mixture was transferred to an amber Erlenmeyer flask, and 50 mL of n-hexane was added. The mixture was shaken and allowed to stand (in darkness) for 20–30 min to allow separation. 25 mL of nano-pure water was added and allowed to stand until complete phase separation. An aliquot of carotenoid solution (hexane phase) was transferred into a glass cuvette, and spectrophotometric readings at 470 nm were collected using a JASCO V-530 UV/VIS–Spectrophotometer. A standard curve was prepared using β-carotene as standard (concentration ranging from 0 to 15 mg/L). TCC results were expressed as milligrams of β-carotene equivalents per 100 g (mg βCE/100 g) of SP samples (dry weight basis).

### 2.4. Sensory Evaluation

For the sensory evaluation, 20 trained panel members were selected aged between 24 and 65 years, where 70% were women. Each panelist evaluated five samples of SP samples dehydrated for 200 min at each tested temperature. A 5-point hedonic scale was used to evaluate dried SP samples’ Appearance, Color, Texture (Firmness), Flavor and Overall Acceptability, with anchor 1 corresponding to “disliked a lot” and anchor 5 to “liked a lot”. In accordance with the Dutcosky-described methodology [24], the panelists were additionally asked to comment on their perceptions and “purchase intent” for the samples. The acceptability index of the different dried SP was calculated as AI = A/B × 100, where “A” is the average grade obtained for the product and “B” is the maximum grade given to the product. Dried SP samples were deemed acceptable when AI was at or above 70% [25].

### 2.5. Statistical Analysis

Data from the trial were subjected to analysis of variance (ANOVA) using the Statistica^TM^ v.8 Software from Statsoft [26]. Statistically significant differences (*p* < 0.05) between samples were determined according to Tukey’s test. Pearson correlation coefficients were also generated between the studied responses. A principal component analysis (PCA) was performed to correlate sensory and physical-chemical parameters for all variables at a drying time of 200 min. Prior to analysis, all variables were mean-centred and scaled to unit variance for principal component analysis (PCA) (correlation matrix). The research data correlation matrix’s eigenvalues and eigenvectors were computed to produce the principal components [27] A score for each sample was calculated as a linear combination for each quality variable for each PCA component. The parameter loading for the factor was used to determine the contribution of each variable to the PCA score. A bi-dimensional representation of this multi-dimensional dataset was made for the primary components that accumulated a significant amount of original information, above 70%, which is deemed sufficient to establish a good model for qualitative purposes [28]. A hierarchical cluster analysis was performed; the clustering procedure included data standardization, evaluation of a sample dissimilarity metric, and application of a grouping technique. The grouping method was Ward’s approach, and the Euclidean distance served as the dissimilarity distance.

## 3. Results and Discussion

### 3.1. Effects of Drying Conditions on the Quality of Dried Sweet Potato

#### 3.1.1. Drying Process and Moisture Content

The drying curves expressed by the evolution of mass loss (g) until arriving at a constant value are shown in Figure 1a. Mass loss (ML) curves show a faster rate within the first 120 min, which could be considered the steady-rate drying period, where the evaporation is set at a uniform rate and water from the inside of the material moves to the surface by diffusion, followed by a slowing down in the subsequent drying period, where it reached the falling rate drying period. Furthermore, data demonstrate that mass loss in SP samples increases as temperature rises, reaching values of about 85% at 80 °C. Similar behaviour was characterized by Gupta et al. [29] during the drying of cauliflower in a thin layer of a convective drier at three temperatures of 55°, 60° and 65 °C with different velocities.

The effect of drying temperature on moisture content (MC; %) of SP samples during time is shown in Figure 1b. The initial moisture content of SP samples was 81.8 ± 0.6%, and samples were dried up to a moisture content ≤10%. As expected, increases in operating temperature led to a faster decrease and lower values in the MC of SP samples.

The ML and MC curves do not reflect precisely the same water loss behaviour. The first curve (Figure 1a) shows a constant rate after 120 min, which is not observed in the second (Figure 1b). The differences pointed out are supported by the different methodologies for evaluation of the water content during the drying process, in which the MC (%) is the most accurate.

#### 3.1.2. Shrinkage and SP Shape

As the amount of liquid water in the porous structure decreases during drying, the capillary pressure increases and causes mechanical stresses in the solid matrix. The natural reaction of the solid matrix is to change its pore space, leading to product shrinkage [30,31,32]. The consequent change in sample shape might affect other product quality attributes such as color, flavour and aroma [32].

Figure 2 shows the samples’ shrinkage values (%) under the tested drying conditions and the photographic records of the SP samples dried for 200 min at each temperature. For all tested temperatures at 20, 60 and 120 min, a significant increase in the shrinkage values of the samples could be observed coinciding with higher moisture content loss rates of the samples (Figure 1b). Maximum shrinkage values of c.a. 48% for all samples are obtained at 200 min. The extent of shrinkage strongly depends on matrix mobility; in particular, it was more noteworthy during the constant and the falling rate periods, when matrix mobility is higher [33,34]. The drying temperature did not significantly influence the shrinkage values of the SP samples. Yadollahinia and Jahangiri [34] also observed that temperature did not affect the shrinkage of potato slices dried at temperatures of 60°, 70°, or 80 °C. However, most of the investigations into dried fruits and vegetables has shown that temperature higher than 70 °C significantly affects material shrinkage [35].

#### 3.1.3. CIELab Color

The effects of the drying conditions (ΔT = 60–80 °C; Δt = 0–200 min) on the CIELab color parameters of the SP samples (C*, °h, WI and ΔE) are presented in Figure 3 (a, b, c and d, respectively).

A significant change occurred in the CIELab color parameters of the dried SP compared to the raw vegetable (time 0). The general trend points to a decrease in the mean °h and C* values (more orange hue and loss of color intensity trends) and an increase in WI (discoloration trend). Overall, during the drying process, SP samples’ color was significantly affected (*p* < 0.05), regardless of the time and temperature applied. However, the time variable was much more important than the temperature variable (F values of 109.7 vs. 6.0, respectively).

Regardless of the temperature (except for 80 °C/20 min), the C* values in the samples denote no additional variations between drying times of 5 to 60 min (Figure 3a). However, for more extended periods (at 120 and 200 min), there is a further tendency for the average chromaticity values to decrease.

For hue angle (°h), the subsequent variations up to 200 min (Figure 3b) denoted no significant changes regardless of temperature and drying times, with only a few exceptions, noted at 200 min for the highest drying temperatures (70°, 75° and 80 °C). Under these conditions, the °h-values returned to the values characterized by the raw material.

The WI index variation (Figure 3c) remains unchanged in the periods from 5 to 60 min, regardless of the drying temperature, except for the highest temperature (80 °C). At longer times (120 and 200 min), the sample’s WI values tend to increase (discoloration trend) and are similar independently of the temperature. Drying SP samples at 80 °C significantly prevented product discoloration (lower WI) during these periods.

The total color differences (ΔE; Figure 3d), irrespective of the drying temperature, reveal significant changes of SP from 5 min of drying (ΔE > 6) maintained until 60 min, with increases (ΔE > 10) in the two tested periods (120 and 200 min).

Grabowski et al. [36] report that the loss of the total amount of beta-carotene in sweet potatoes during thermal processing, such as hot air drying, is caused by the isomerization of carotenoids by heat. Other factors, such as temperature, pH, aw and sweet potato cultivar, can also affect browning changes [37,38]. The existing literature states that color changes due to the drying process could be explained by different mechanisms, namely through enzymatic browning reactions, Maillard reactions or pigment oxidation [39]. As mentioned in [40] regarding orange sweet potatoes, the high levels of carotenoids, as in the variety under study, may confer some level of protection to these degrading mechanisms, namely those responsible for browning (as shown by the parameter °h). The hypothesis that the high TCC confers some protective factor may be attributed to the levels of carotenoids acting as antioxidants and thus preventing the oxidation of polyphenols, which are responsible for browning reactions. In drying processes, the formation of antioxidant compounds originating in the Maillard reactions could also protect the carotenoids from oxidation [41].

#### 3.1.4. Texture Properties

The force–distance curves obtained for SP samples dried for 120 and 200 min are typical of crispy foods and indicate the phenomenon known as “brittle fracture” [42]. The samples’ initial peak, or maximal strength, corresponds to their hardness depending on the substance’s mechanical characteristics and the structure’s components (air cells and geometry). The behaviour of a brittle object reveals high hardness (H), little work (W) until fracture (BD), and a sudden drop in strength as the crack propagates rapidly [43]. Figure 4a–d shows the texture properties, hardness (H), breaking distance (BD), work of fracture (W), and crispness (CP), respectively, of SP samples. Again, the textural properties evaluated depended more on time than drying temperature (F values of 126.13 vs. 10.16, respectively). The hardness (H) of the SP samples changed significantly from the initial values (t = 0) only for periods longer than 60 min (except for 80 °C at 60 min) (Figure 4a). At longer drying periods (≥120 min), the increases in the samples’ H proved to be independent of temperature. The results of Rashid et al. [44] concerning the drying of sweet potato demonstrated the achievement of a desirable texture (hardness and resilience) achieved for a drying temperature of 70 °C, slightly lower than our results (requiring higher values greater than or equal to 75 °C). However, in the study of Rashid et al. [44], the drying was performed in hot air in combination with ultrasound, and the synergistic effects developed promoted the rate of water loss.

The samples’ BD and W profiles (Figure 4b,c) are similar over the different drying times tested. This verified the maintenance of the initial values (until 20 min), independently of the temperature. At 60 min, increments (*p* < 0.05) are observed with increasing temperature (except ≥ 75 °C). For the longer drying periods of 120 and 200 min, the inverse behaviour is denoted: a decrease of the respective BD and W values as the temperature increases.

The samples’ crispiness property (CP) showed significant increases (Figure 4d) only in the more extended drying periods. Proportional increases were observed with increasing temperature (significantly higher values at 75° and 80 °C at 120 and 200 min). Crispness is a crucial textural property in dry products, contributing significantly to their palatability and dependent on the moisture content. Our results show that this property was achieved when the samples reached less than 10% moisture content.

#### 3.1.5. Total Phenolic (TPC) and Total Carotenoid (TCC) Contents

The TPC results for the raw SP, 189.01 ± 9.8 mg GAE/100 g dw, were within the range of values for different sweet potato varieties, ranging from 192.7 to 1159.0 mg GAE/100 g dry basis [45].

The SP samples’ TPC changes under the different tested drying conditions, shown in Figure 5a, indicate a significant effect on the loss of total phenolic content compared to the raw content under the influence of time rather than the applied temperature (F values of 221.18 vs. 47.25, respectively). Although there is a tendency to exacerbate phenolic losses with the increase in drying temperature, the losses increase significantly with the drying time, independently of the temperature. In fact, at t = 200 min for all temperatures, the loss of total phenolics compared to the raw content reaches values of about 70%.

High TPC losses (>40%) are commonly reported for various heat-dried fruits and vegetables, including sliced tomatoes [46], cut Galega Kale leaves [47], broccoli florets [48], and cut pomegranates [49], regardless of the tested conditions. However, under a different time–temperature regime (high temperatures, short times) compared to our investigation, the study by Galaz et al. [50] on pomegranate peel showed that fresh fruit TPC levels were maintained after drying at high temperatures (100°, 110°, and 120 °C).

The phenolic loss induced by conductive drying processes can be ascribed to enzymatic decomposition, development of insoluble oxidation products, polymer production, and thermal degradation of phenolic acids and catechins [49,51]. Thermal stability also depends on the type of phenolic compounds involved, altering the phenolic profile of the dried product [52,53]. On the other hand, in the food matrix, cellular structures and the binding of phenolic compounds to other macromolecules (such as proteins or polymeric carbohydrates) can be broken down by heat treatment, favouring the release of phenolic compounds leading to increases in TPC after drying [52,54]. Therefore, the thermal stability of phenolic compounds is highly dependent on the product type, its anatomical structure, and the chemical properties of the bioactive compounds [54].

The drying method also influences the extent of phenolic degradation. The drying study of sweet potato slices by Yang et al. [55] showed that hot air drying provided a higher phenolic loss when compared to the microwave method. Finally, some authors also refer that the quantification of TPC values after dehydration may still be underestimated by the currently available methods, taking into account the interactions of polyphenols with proteins or changes in their chemical structure [56,57]. On the other hand, reducing sugars may also contribute to overestimating the TPC results [58]. Since the increase of sugars under concentration effect during the drying is expected, the TPC losses characterized in the SP samples may have been even higher.

The total carotenoid content (TCC) for the raw ‘Bellevue’ SP, of 7.6 ± 0.2 mg βCE/100 g fw (41.6 ± 3.2 mg βCE/100 g dw), was within the range for different sweet potato varieties, i.e., from 7.5 to 15.5 mg βCE/100 g fw [59]. The TCC assessed in the dried SP samples under the different drying conditions is shown in Figure 5b, and can be viewed as a quality index for dried orange products, expressing the changes in color pigments and nutritional value [60]. It should be noted that the sample’s TCC changes under the tested conditions were much more moderate than the changes in the total phenolic content, never exceeding 30%. For TCC changes, the temperature had a more significant influence than drying time (F values of 97.89 vs. 46.28, respectively).

As a general trend, the loss of carotenoids correlated inversely with the progressive increase in drying temperature. On the other hand, TCC decreased as drying time increased, except for temperatures of 75° and 80 °C. Under these conditions, the TCC of the SP samples for all tested periods retained the content of the raw vegetable (41.6 ± 3.2 mg βCE/100 g dw).

In a hot air drying study of apricots, Ihns et al. [61] found similar behaviour by observing that the β-carotene content of two fruit varieties increased with increasing drying temperature (80° and 100 °C). This behaviour was attributed to the increased solubility of β-carotene with increasing temperature. They also found that fruits dried for longer times resulted in lower β-carotene contents, justified by a greater opportunity for oxidation. Thus, apricots dried for longer periods and at lower temperatures resulted in lower β-carotene contents.

Retention of carotenoids could also be related to the morphology of chromoplasts, containing pigments as carotenoid sources. Zhang et al.’s study [62] compared the bio-accessibility of carotenoids from different vegetables (carrot, sweet potato, yellow pepper and broccoli) during hot air drying at different temperatures. Drying led to cell wall rupture, inducing the release and loss of carotenoids from plant products. On the other hand, starch granules, as structural barriers, have been identified as essential for protecting carotenoids in sweet potatoes [62].

### 3.2. Effects of Drying Temperature on the Quality of Sweet Potato Dried at 200 min

#### 3.2.1. Sensory Evaluation

In the last period tested (t = 200 min), the SP samples dried at different temperature conditions were evaluated by a sensory panel. The mean ratings obtained on the different attributes (Appearance, Color, Texture, and Flavor) are shown in Figure 6.

Slight differences (*p* > 0.05) between SP samples dried at different temperatures were found in Appearance and Color attributes. The highest Texture and Flavor (Aroma/Taste) scores were obtained for samples dried at higher temperatures (70°–80 °C) (Figure 6), with significant differences registered only for samples dried at 60 °C. The overall acceptability ratings showed a preference for samples dried at high temperatures (>70 °C). On the contrary, the SP samples dehydrated at 60 °C were sensorily rejected, obtaining ratings < 3 (midpoint of the scale). The dehydrated samples at 75 °C obtained the highest AI value of 80%, and 97% of the panelists expressed interest in buying this product, primarily due to its taste and convenience.

The sensory ratings of color and appearance attributed to the samples showed a correlation (0.90, *p* < 0.05). However, only the sensory ratings for texture and flavor were determinant in the overall appreciation of the SP samples (significant correlations between the ratings of 0.99 and 0.98, respectively). These correlations were to be expected since the drying temperature did not significantly influence color between samples (for t = 200 min).

#### 3.2.2. PCA Modelling

In order to investigate how drying temperatures (for t = 200 min) affected overall SP quality parameters, including sensory judgements, we used a principal component analysis (PCA). A preliminary PCA was performed to determine which (quantitative) variables contributed to the model. According to low factor loadings, the variables Shrinkage, Appearance_S, Color_S, and WI were excluded from the model. The variable W was eliminated as it was autocorrelated with BD. Hierarchical cluster and principal component analyses were used to establish a relationship between the drying temperature and SP quality attributes. The resulting data matrix contained 15 samples and 12 variables. The data set included five categories of samples, identified by the dry temperatures (60°, 65°, 70°, 75° and 80 °C). The quantitative variables corresponded to C*, °h, deltaE, MC, H, BD, CP, TPC, TCC, Text_S, Flav_S and OAP_S.

The PCA explained 70.99% of the original data variability in the first two dimensions (Figure 7). PC1 accounted for the data’s most significant proportion of variation (49.02%), and it was most loaded with the textural properties (BD and CP), the TCC, the moisture content (MC), and the sensory ratings (Text_S, Flav_S and OAp_S) variables. It is noted that the distribution of the samples in the score plot (Figure 7b) shows the increase in drying temperature from right to left along PC1, helping to interpret the effect of temperature on product quality.

In Figure 7a, the vector projection of the variable PC1 reveals that favourable sensory appreciation (Text_S, Flav_S and OAp_S) significantly depended on increasing sample crispness (CP) resulting from higher drying temperature, as all these variables correlate negatively with PC1. On the other hand, the inverse correlation of the CP and BD variables with PC1 shows the dependence of these texture properties on dry products. The higher the distance to break off the sample, the lower the level of the food’s crispness. The projection of the TCC variable onto PC1 (significantly inverse correlation) supports the previously mentioned trend between higher drying temperatures (75° to 80 °C) and higher sample’s carotenoid content. The opposite correlation of TCC and moisture content (MC) variables with PC1 reveals a concentration effect. Additionally, it was found that the inverse correlation of all sensory variables and MC along PC1 reveals that lower MC values (ca 10%) leads to high sensory acceptance, only reached for drying temperatures greater than 75 °C (Figure 1b).

PC2 (which explains 21.98% of the data variability) describes the dynamics of color change between samples at different temperatures (Figure 7a). It is represented only by deltaE variable negatively correlated with PC2. This indicates that samples’ total color changes, regardless of temperature, did not influence the SP’s sensory acceptance. This conclusion agrees with the results of the color sensory appreciation of the samples (t = 200 min) (Figure 6), whose ratings showed no significant differences (t = 200 min), regardless of drying temperature.

The score plot (Figure 7b) indicates sample grouping as revealed by the hierarchical cluster analysis dendrogram (Figure 8). According to sample clustering, the first division into two groups over PC1 separates the samples dried at 60 °C from the others. As a result, the conclusion that this temperature does not ensure SP chips with acceptable sensory and bioactive quality is also supported.

## 4. Conclusions

The effect of air drying temperature (from 60° to 80 °C) on the quality attributes (physical-chemical, bioactive and sensorial attributes) of SP chips (Belleveu PBR; 2.0 ± 0.1 mm) was investigated. In the SP drying process, processing time proved to be the most determinant variable, over temperature, in changing all evaluated quality parameters except carotenoid content.

The retention of carotenoids was ensured despite the higher temperatures applied (75° and 80 °C), possibly because of the structure of the tuber, containing starch granules, which offer protection from thermal degradation. On the contrary, the thermal degradation of phenolics induced by drying was high, reaching values of about 70% (t = 200 min), regardless of temperature.

Furthermore, to achieve moisture values that guarantee the dry product stability (<10%), it was necessary to apply high drying times (>120 min) combined with high-temperature values (≥75 °C). In this study, temperature values ≥ 75 °C for times of 200 min allowed the production of sweet potato chips with appreciated sensory quality (flavor, color, texture), crispness values of >4 N/mm and total carotenoid content (≈42 mg βCE/100 g dw) equivalent to the raw product.

To provide healthier alternatives for the increased intake of vegetables, which is advised in the human diet as a source of fiber and carotenoids, hot air drying has been optimized as a simple and cost for value production process for non-fried sweet potato snacks.

## Figures and Tables

**Figure 1 foods-12-01198-f001:**
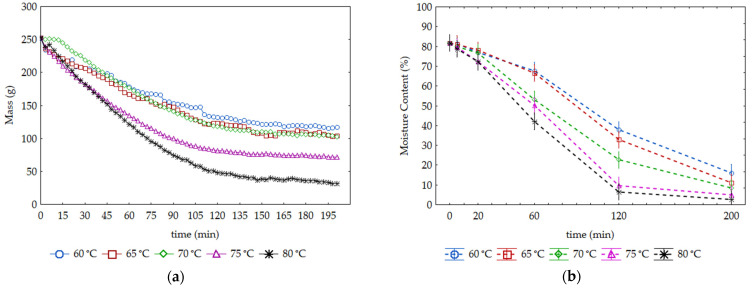
Effect of drying temperature on mass loss (g) (**a**) and variation of moisture content (%) (**b**) of SP samples with time. Vertical bars denote 0.95 confidence intervals.

**Figure 2 foods-12-01198-f002:**
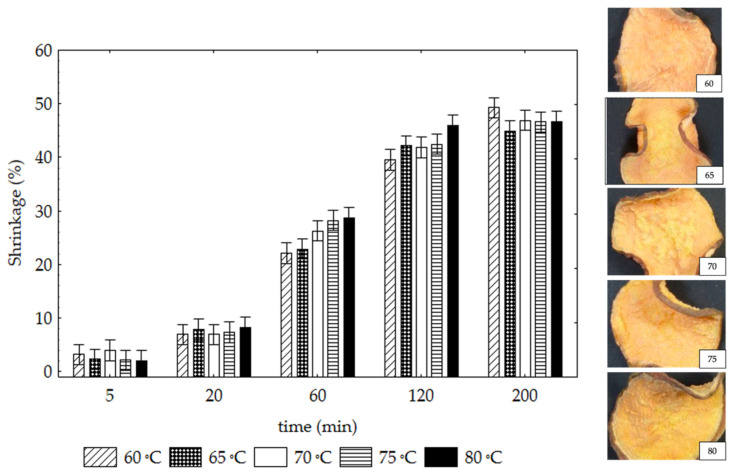
Drying process effect on shrinkage values (%) of SP samples at different temperature-time conditions and photos of the dehydrated product at the end of 200 min. Vertical bars denote 0.95 confidence intervals.

**Figure 3 foods-12-01198-f003:**
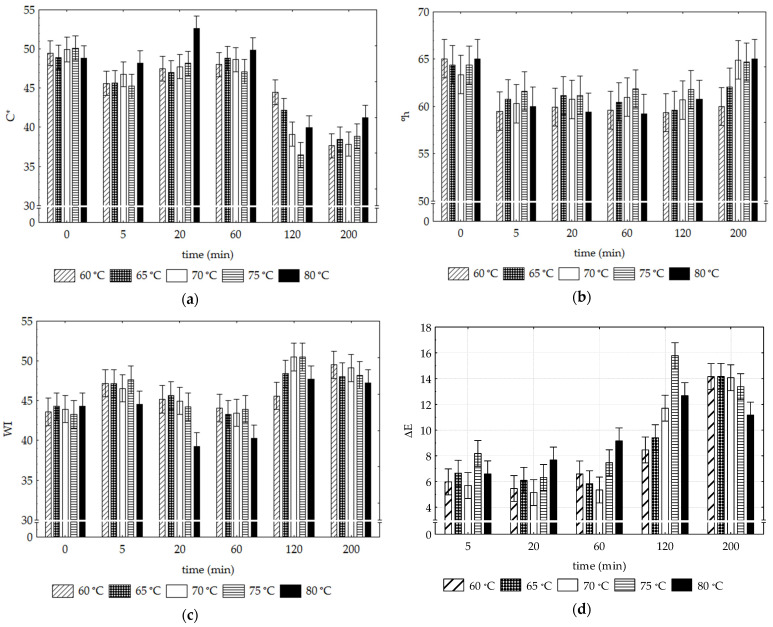
SP samples’ color indices change as a function of temperature and time of drying: (**a**) C* (Chroma), (**b**) °h (Hue angle), (**c**) WI (Whiteness Index) and (**d**) ΔE (Total Color Difference). Vertical bars denote 0.95 confidence intervals.

**Figure 4 foods-12-01198-f004:**
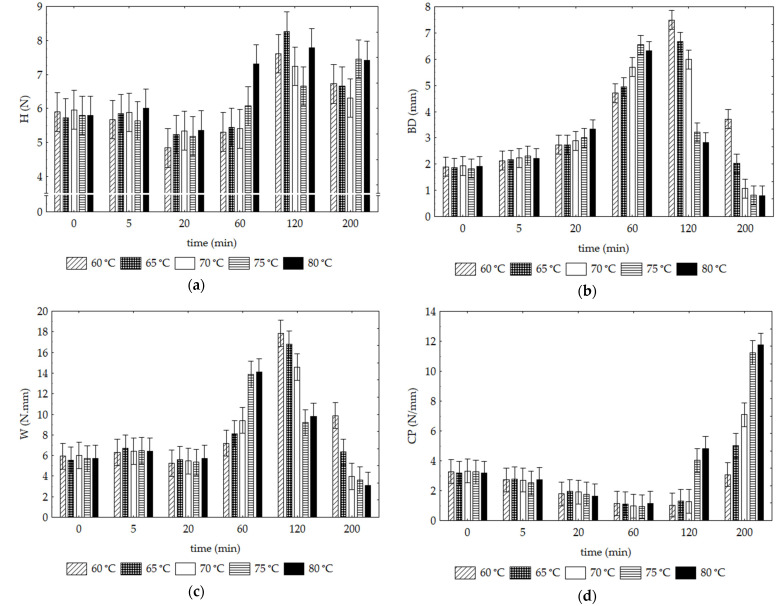
Sweet potato texture parameters change as a function of temperature and time of drying: (**a**) hardness (H), (**b**) breaking distance (BD), (**c**) work of fracture (W) and (**d**) crispness (CP). Vertical bars denote 0.95 confidence intervals.

**Figure 5 foods-12-01198-f005:**
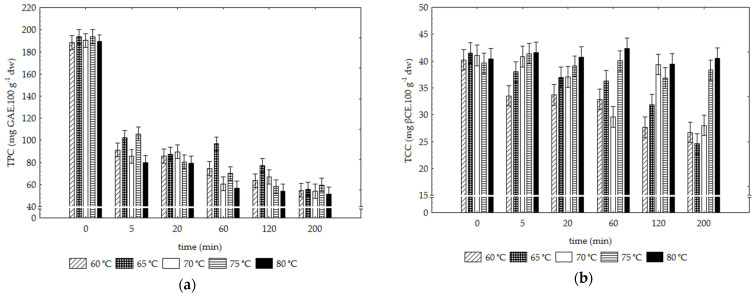
Sweet potato TPC (**a**) and TCC (**b**) changes in the function of temperature and time of drying. Vertical bars denote 0.95 confidence intervals.

**Figure 6 foods-12-01198-f006:**
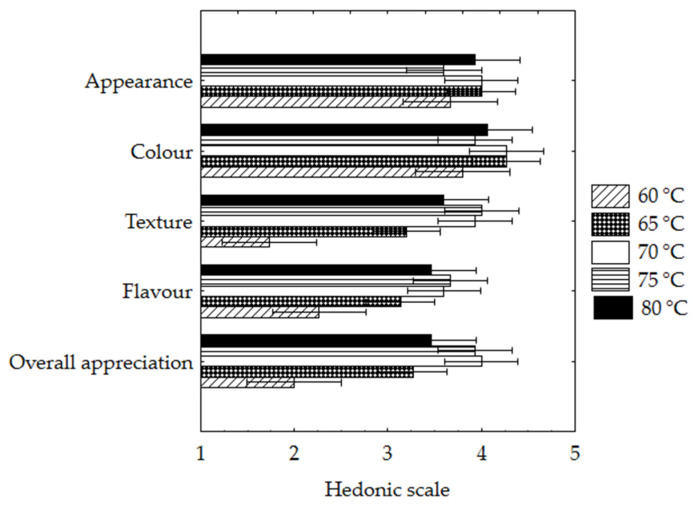
Sensory scores for the attributes Appearance, Color, Texture, Flavor, and Overall appreciation of dried sweet potatoes at different temperatures, 60 °, 65 °, 70 °, 75 ° and 80 °C, at 200 min.

**Figure 7 foods-12-01198-f007:**
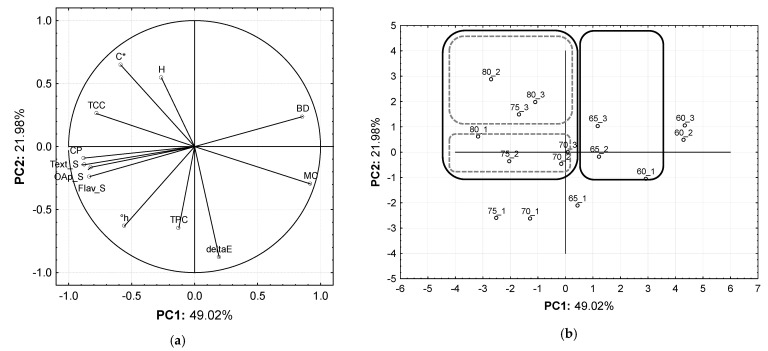
Principal component analysis (PCA) of SP samples as affected by different dried temperatures (60°, 65°, 70°, 75° and 80 °C) at 200 min: (**a**) score plot and (**b**) loading plot. C*–Chroma, °h–hue angle, deltaE–Total Color Difference, MC–Moisture Content, H–Hardness, BD–Breaking Distance, CP–Crispness, TPC–Total Phenolic Content, TCC–Total Carotenoid Content, Text_S–Texture (sensorial), Flav_S–Flavor (sensorial) and OAP_S–Overall Appreciation (sensorial).

**Figure 8 foods-12-01198-f008:**
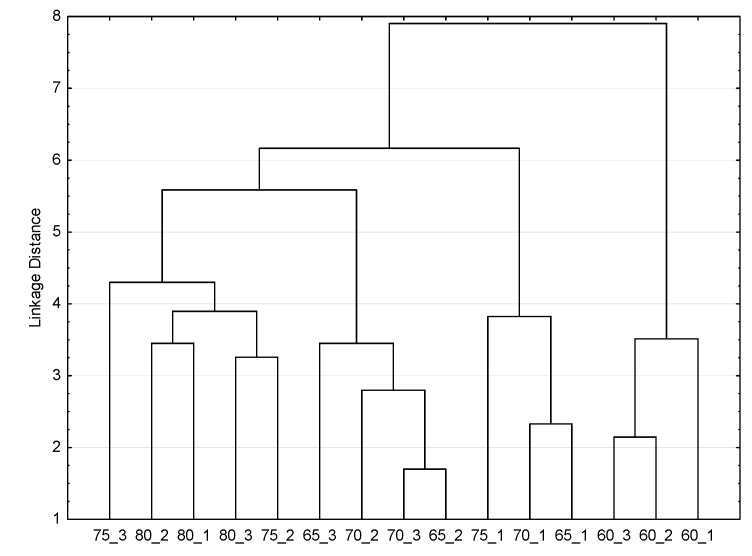
Hierarchical cluster analysis dendrogram of the data matrix.

**Table 1 foods-12-01198-t001:** Physical, chemical and phytochemical characterization of raw samples.

Quality Attribute	Mean ± SD *
pH	6.03 ± 0.5
a_w_	0.96 ± 0.01
Soluble solids content (°Brix)	7.1 ± 0.4
Moisture content (MC; %)	81.8 ± 0.6
Titratable acidity (g citric acid/100 g)	0.02 ± 0.01
CIELab Color	
L*	73.0 ± 3.8
a*	21.6 ± 8.7
b*	44.1 ± 5.7
Texture	
Hardness (N)	6.1 ± 1.4
Breaking distance (mm)	1.9 ± 0.7
Crispness (N/mm)	3.3 ± 0.8
Work of fracture (N.mm)	5.6 ± 1.6
Total phenolic content	
(mg gallic acid eq./100 g dw)	189.01 ± 9.8
Total carotenoid content	
(mg β-carotene eq./100 g dw)	41.6 ± 3.2

* Data are means of three replicates per batch.

## Data Availability

Data will be made available upon reasonable request to the corresponding authors.

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
