# Peer review of "Influence of Air-Drying Conditions on Quality, Bioactive Composition and Sensorial Attributes of Sweet Potato Chips"

_foods, 2023, doi:10.3390/foods12061198_

Round 1
Reviewer 1 Report
Foods-2258668
In the submitted article entitled “Influence of air-drying conditions on quality, bioactive composition and sensorial attributes of sweet potato chips” the authors examined the effect of temperature and drying time on the quality of sweet potato chips. Convection drying at 60°, 65°, 70°, 75°, and 80 °C was used, in the range of 5 to 200 min. The basic physical and chemical parameters of the dried material were examined: dry matter content, color, texture, total polyphenol content, total carotenoid content, and sensory evaluation was carried out.
Please refer to the following comments:
Line 81: add storage conditions: temperature and relative humidity,
Line 87: what was the weighing frequency?
Line 110: provide measurement parameters: light source, measurement geometry and type of observer,
Line 125-131: give details for the analysis of total polyphenols and total carotenoids,
Line 131: present the results for both dried material and fresh material per 100 g of dry substance,
For Figure 4: For hardness add a legend.
Line 297: How to explain the increase in the content of total polyphenols in the material after 60 minutes of drying at 65 C?
For Figure 5: Add values for time 0 (raw material).
Line 345: What can explain the increase in better sensory parameters for samples dried at higher temperatures (70,75 C)?
In the discussion (the content of bioactive compounds), the authors should take into account the imperfection of the Folin-Ciocalteu method. The F-C reagent also reacts with reducing sugars and vitamin C present in the material. The changes in physical and chemical parameters occurring during the convective drying process should be tried to be explained in a broader way, especially since the subject was taken up by other scientists.

Reviewer 2 Report
This manuscript addresses air-drying conditions of sweet potato chips.. The authors analyze physicochemical values, contents of polyphenols and beta-carotene and sensory evaluation. Then, these provide some useful information for optimization of the production processes, however some minor revisions should be needed, especially discussion in sensory analysis.
l Please discuss the results of sensory evaluation using physicochemical data. For example, the author illustrated lowest value of flavor, texture and acceptance in sample 60C and then please describe the reasons based on physicochemical data. Also, please re-write Figure 6 (a) because it is very difficult to recognize samples. Moreover, it was observed that sample 80C showed higher score in Figure 6 (a), but the score in preference test was not highest. Please demonstrate these results.
l Which condition is best for commercial production, including liking/disliking, the level of polyphenols and beta-carotene remained and effective drying? Please show your hypothesis based on this research.
Reviewer 3 Report
The submitted manuscript under ID foods-2258668, is dedicated to evaluation of air drying temperature and time on the quality properties (bioactive, sensorial and physical-chemical) of sweet potato chips. The presentation, structure and analysis of the work are appropriate, but there is a lack of originality and innovation.
Specific comments regarding the work are as follows:
Introduction: Previous work done in this field as well as novelty and objectives of this study is not clearly described. There are already studies dealing with sweet potato chips and the effect of air-drying, what is new about this study?
In general, mention and compare some other types of drying on the quality of sweet potato chips to highlight the advantage of this type of drying.
Figure 1. Mass loss curves (g) (a) and variation of moisture content (%) (b) of SP samples with time. Add depending on the temperature in this figure description. Temperature markings in the form of stars, squares, etc. that are unclear in the figures and should be marked differently, for example with different colors.
Line 238-239. An explanation of how you came to the conclusion that carotenoids affect the protection of browning and Maillard reactions?
Line 244-266 Compare the obtained results with the results of testing the texture of sweet potato chips by other authors.
Line 263-265 Reference missing.
Line 330-353 Why the effect of drying time was not taken into account in the sensory evaluation?
Table 2. unnecessary to show, it is better to show a dendogram (Figure 7b).
Conclusion: Why it is claimed that these chips are a healthy alternative to vegetable consumption (when a lot of phenolic compounds are lost). Clarify and better emphasize the purpose of this and the proposal for further research.
Round 2
Reviewer 3 Report
The manuscript is sufficiently improved to warrant publication in Foods.